# Expression site attenuation mechanistically links antigenic variation and development in *Trypanosoma brucei*

Christopher Batram, Nicola G Jones, Christian J Janzen, Sebastian M Markert, Markus Engstler*

Department of Cell and Developmental Biology, University of Würzburg, Würzburg, Germany

**Abstract** We have discovered a new mechanism of monoallelic gene expression that links antigenic variation, cell cycle, and development in the model parasite *Trypanosoma brucei*. African trypanosomes possess hundreds of *variant surface glycoprotein* (*VSG*) genes, but only one is expressed from a telomeric expression site (ES) at any given time. We found that the expression of a second *VSG* alone is sufficient to silence the active *VSG* gene and directionally attenuate the ES by disruptor of telomeric silencing-1B (DOT1B)-mediated histone methylation. Three conserved expression-site-associated genes (ESAGs) appear to serve as signal for ES attenuation. Their depletion causes G1-phase dormancy and reversible initiation of the slender-to-stumpy differentiation pathway. ES-attenuated slender bloodstream trypanosomes gain full developmental competence for transformation to the tsetse fly stage. This surprising connection between antigenic variation and developmental progression provides an unexpected point of attack against the deadly sleeping sickness.

*For correspondence: markus.engstler@biozentrum.uni-wuerzburg.de

Competing interests: The authors declare that no competing interests exist.

## Introduction

Functional variation between cells in a population is often achieved by selective expression of one protein from a pool of possibilities. To achieve this goal, a variety of mechanisms have evolved that ensure allelic exclusion, namely the silencing of expression of all but one member of a gene family, either temporarily or for the remainder of the life of the cell.

In the mammalian central nervous system, each olfactory sensory neuron expresses only one olfactory receptor (OR) from a family of ~1200 genes (*Buck and Axel, 1991*). Before an *OR* gene is expressed, all alleles are silenced and converted to heterochromatin (*Magklara et al., 2011*). A limiting enzymatic activity then stochastically removes the heterochromatin marks from one allele to activate it. The expressed OR protein mediates a feedback loop that inhibits removal of hetero-chromatin marks from all other alleles, preventing their transcription (*Serizawa et al., 2003*; *Lyons et al., 2013*).

Allelic exclusion commonly occurs in pathogens that exploit antigenic variation of their cell surface proteins to keep ahead of the host immune response, usually as part of a population survival strategy. The malaria parasite *Plasmodium falciparum*, for example, expresses only one out of 60 members of the *var* gene family, each coding for different versions of the surface virulence factor PfEMP1 (*Guizetti and Scherf, 2013*).

Monoallelic expression of the variant surface glycoprotein (VSG) in *Trypanosoma brucei* is a particularly striking example of allelic exclusion in a pathogen. In the mammalian host, the cell surface is covered with millions of copies of VSG that form a dense layer that is virtually impervious to host-derived anti-bodies (*Cross, 1975*; *Engstler et al., 2007*; *Schwede et al., 2011*). VSGs are highly immunogenic

**eLife digest** African sleeping sickness is a potentially lethal disease that is caused by a parasite called *T. brucei* and spread by tsetse flies. Like many of the parasites that cause tropical diseases, *T. brucei* employs genetic trickery to evade the immune systems of humans and other mammals. This involves changing the variant surface glycoprotein (VSG) coat that surrounds the parasite on a regular basis in order to remain one step ahead of the immune system of its host: while the immune system looks for invaders wearing a particular coat, the parasites are spreading through the host in a completely different coat.

To infect other hosts, the parasite must undergo changes that allow it to re-infect the tsetse fly. Therefore, besides the 'antigenic variation' that allows *it* to change its surface coat when it is in the blood of its host, *T. brucei* must undergo a more fundamental metamorphosis before it is capable of colonizing the tsetse fly. However, many details of the changes that allow the parasites to re-infect flies are not understood.

*T. brucei* has several hundred VSG genes clustered in about 15 regions known as expression sites, but only a single expression site is active at any given time. Each expression site also contains a number of other genes known as expression site-associated genes (ESAGs). Antigenic variation can occur as a result of different VSG genes within the same expression site being expressed as proteins, or when the active expression site is silenced and another expression site is activated. This is another process that is not fully understood.

Batram et al. now reveal that the expression of VSG genes, antigenic variation and the changes that allow the parasites to re-infect flies are all related to each other. This suggests that the expression site could provide a new point of attack in the fight against African sleeping sickness.

and provoke a rapid and efficient immune response that diminishes the parasite population. Only trypanosomes that have successfully switched to expression of another, structurally similar but immunologically distinct VSG survive. Trypanosomes possess several hundred *VSG* genes (*Berriman et al., 2005*). Their potentially unlimited capacity for antigenic variation forms the basis of trypanosome persistence and virulence. *VSG* genes are expressed from one of ~15 telomeric expression sites (ES) but only one of these is transcribed at any given time (*Hertz-Fowler et al., 2008*). Chromatin remodeling appears to play an important role in maintaining the monoallelic expression of the active ES (*Horn and McCulloch, 2010*). Antigenic variation can result from either a change in the active ES, usually gene conversion at the *VSG* locus, or by an epigenetic change that results in silencing of the active ES and transcription of a previously silent ES (in situ switch). No mechanism or factor has been identified that is involved in the initiation of the latter.

The complex life cycle of *T. brucei* represents a succession of proliferative and quiescent developmental forms, which vary widely in cell architecture and function (*MacGregor et al., 2012*). Throughout the parasite's life cycle, the plasma membrane is covered with a series of different surface coats. Antigenic variation only occurs in the mammalian host, and not in the transmitting tsetse fly. Consequently, the trypanosome bloodstream form ES is exclusively functional in the proliferating slender bloodstream stage of the parasite. In response to quorum sensing, the trypanosomes differentiate to the quiescent stumpy form. Stumpy forms are competent for the next developmental transition to procyclic forms, which occurs after ingestion by a tsetse fly (*Reuner et al., 1997*). This developmental transition is accompanied by silencing of the active ES so that allelic exclusion is lost. How this silencing is achieved is unknown.

In this study, we show that high-level expression of an ectopic *VSG* transgene is sufficient to initiate the directional silencing of the active ES from the telomere inwards. The silencing of the active ES is dependent on the histone methyltransferase DOT1B. Thus, the VSG itself mediates a feedback loop that controls the state of monoallelic gene expression. Furthermore, we show that transcriptional activity of the ES directly determines cell proliferation and developmental competence via a subset of *expression site associated genes (ESAGs)*. Thus, the trypanosome ES can be regarded as a fine-tuned regulator at the crossroads of antigenic variation, proliferation, and development.

## Results

### Expression of wild-type levels of a second VSG causes gradual attenuation of the active expression site

To simulate a transcriptional switching event (in situ switch) in the most straightforward manner, we adopted a strategy that allows inducible high-level expression of *VSG121* in trypanosomes that have *VSG221* in their active expression site (221$^{ES}$). This was achieved by introducing a *VSG121* gene under the control of a tetracycline-inducible T7 RNA polymerase promoter (121$^{tet}$) into an rDNA spacer locus (*Figure 1A*). We decided to use the T7 RNA polymerase-based expression system instead of a polymerase I-based system to minimize interference with the endogenous transcription machinery. Quantitative Northern blot analysis revealed an almost instantaneous increase of *VSG121* mRNA, reaching 82(±1)% of wild type levels within 2 hr (*Figure 1B*). This was followed by a reduction of endogenous *VSG221* transcripts, which declined with kinetics compatible with the mRNA half-life (*Ehlers et al., 1987*). The total *VSG* mRNA initially peaked at 180%, but within 8 hr leveled to approximately wild-type quantities. Thus, the *VSG* mRNA population was effectively exchanged in this time period.

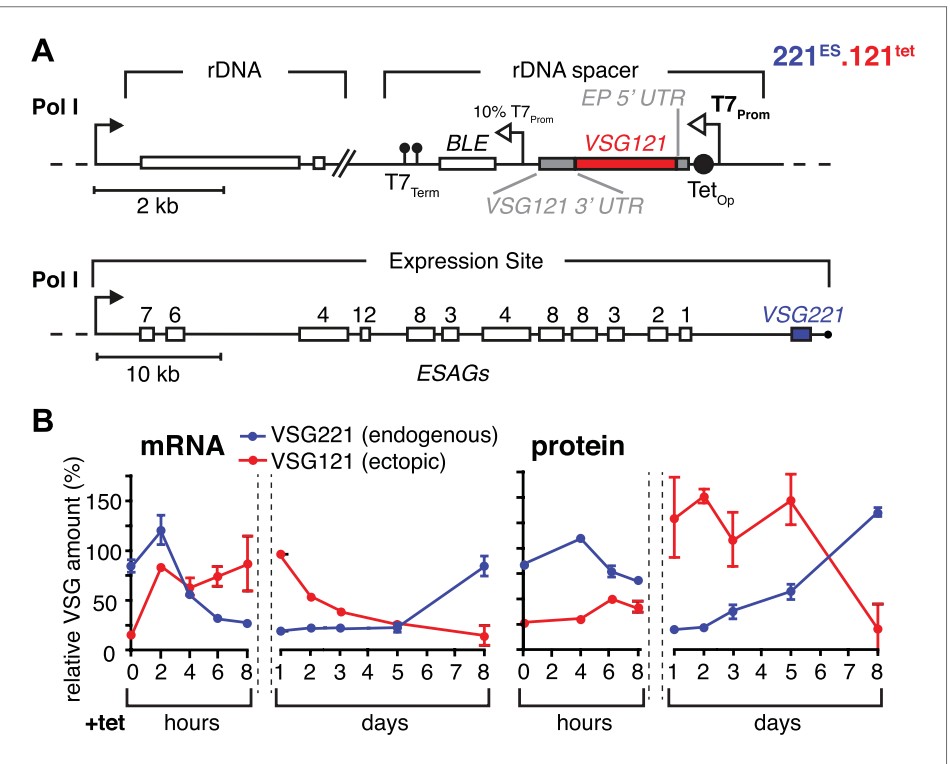

**Figure 1**. Inducible *VSG121* expression leads to endogenous *VSG* silencing. (**A**) Illustration of the ectopic *VSG* expression strategy. *VSG121* integration into the transcriptionally silent rDNA spacer is achieved in the absence of tetracycline by using a constitutive 10% T7 promoter driving the *BLE* resistance cassette. High-level expression of ectopic *VSG121* occurs upon tetracycline induction of a full T7 promoter. (**B**) Quantification of *VSG* mRNA and protein levels during the course of tetracycline-induced *VSG121* expression. The values are percentages ± SD for two independent clones normalized to the parental 221$^{ES}$ or VSG121 wild-type cells. Total RNA samples were prepared at the time points indicated and analyzed by dot blotting. The blots were hybridized with VSG-specific fluorescent DNA probes and quantified by normalization to *beta-tubulin* mRNA using a Licor Odyssey near infrared scanner. Protein equivalents of $6 \times 10^5$ cells were dot blotted and incubated with anti-VSG121 or anti-VSG221 antibodies. Quantification was done by normalization to the paraflagellar rod (PFR) protein using the Licor Odyssey system.

The following figure supplements are available for figure 1:

**Figure supplement 1**. VSG immunofluorescence after tetracycline-induced *VSG121* expression.

The VSG121 protein was detectable within 4–6 hr after induction (*Figure 1B*). After 24 hr, the surface coat of all cells was dominated by VSG121 (*Figure 1—figure supplement 1*) and the amount of VSG221 had declined to 21(±2)% (*Figure 1B*). Thus, expression of an ectopic VSG at wild-type quantities caused a reduction in the abundance of the ES-resident VSG, which decreased with a half-life of 12 hr that corresponds to the population doubling time within the first 24 hr of induction.

The reduction in steady state levels of *VSG221* mRNA was caused by an active process, which specifically affects the ES-resident *VSG*, either through reduction of transcription or by *VSG* mRNA degradation. To distinguish between these possibilities, we quantified other mRNAs derived from the ES. As transcription of the ES is polycistronic, any transcriptional silencing would also affect genes upstream of the *VSG*. Initially, the levels of three *ESAG* mRNAs were measured over a period of 8 hr after *VSG121* induction (*Figure 2A,B*). It appears that mRNAs from the telomere proximal *ESAG1* and *2* genes decreased faster than mRNA from the telomere distal *ESAG12* gene. To examine whether this effect was the result of different mRNA half-lives, we inserted the same *GFP* transgene either upstream of the *VSG* gene (promoter distal) or next to the ES promoter (promoter proximal) (*Figure 2A*). Upon induction of *VSG121* expression, the promoter proximal *GFP* mRNA decreased with significantly slower kinetics (p<0.018; two-way ANOVA test) than that of the promoter distal *GFP* (*Figure 2C*, left panel), revealing that the kinetics of *GFP* mRNA reduction was determined solely by the position of the *GFP* gene within the active ES. Thus, *VSG121* overexpression caused a gradual loss of ES activity that started at the telomere and propagated towards the *VSG* promoter. This process was completed within one cell division cycle and was maintained for 120 hr (*Figure 1B*). Then the ES was gradually

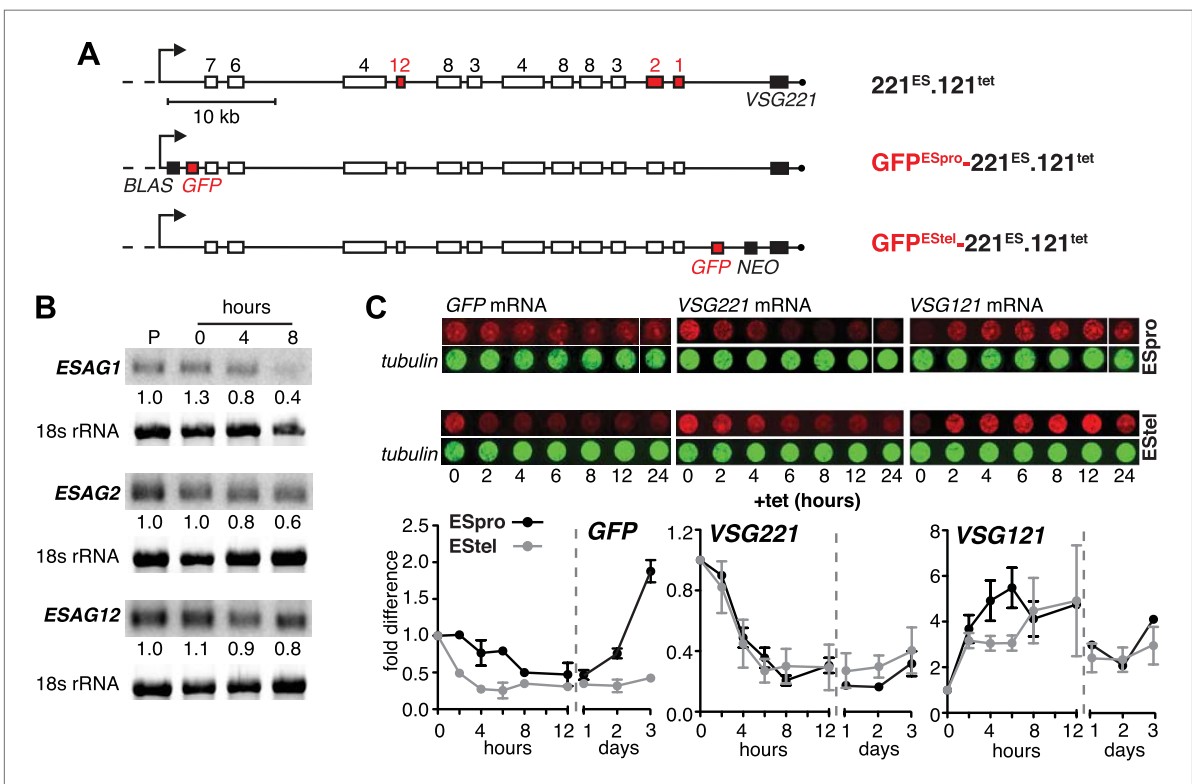

**Figure 2**. Ectopic *VSG121* expression induces gradual ES attenuation. (**A**) Scheme illustrating the anatomy of the *VSG221* expression site in a set of reporter lines. Cell line 221[ES].121[tet] was used to quantify three different *ESAG* transcripts. In GFP[ESpro]-221[ES].121[tet], a *GFP* reporter gene was inserted just downstream of the ES promoter and in GFP[EStel]-221[ES].121[tet] just upstream of the endogenous *VSG221*. *NEO*, neomycin resistance; *BLAS*, blasticidin resistance; numbered boxes, *ESAG*s; arrow, ES promoter. (**B**) *ESAG1* and *2* transcripts decrease faster than *ESAG12* mRNA. After 0, 4, and 8 hr of ectopic *VSG121* induction, mRNA levels of three *ESAG*s were quantified using [32]P-labeled probes on Northern blots and normalized to 18S rRNA using a fluorescent probe. Values are relative to the parental 221[ES] cells (P). (**C**) Expression site attenuation starts at the telomere and is released in the reverse direction. The *GFP* reporter lines were induced for ectopic *VSG121* expression and mRNA levels of *GFP*, *VSG221* and *VSG121* were quantified using fluorescently labeled probes, normalized to *beta-tubulin* mRNA (upper panel). Values are relative to the non-induced levels ± SD for two independent clones (lower panel).

reactivated in the reverse direction (*Figure 2C*). The ES was not silenced completely, so the effect did not constitute a bona fide ES silencing, but a fine-tuned ES attenuation. We postulate that this VSG-induced attenuation represents the initiation step of an antigenic in situ switch.

## Expression site attenuation causes specific prolongation of the G1 cell cycle phase and concomitant global transcriptional diminution

The immediate early effect of *VSG121* overexpression was the attenuation of the active ES (*Figures 1 and 2*). Parasite growth was not affected for the first 24 hr of tetracycline induction, but then cell proliferation slowed down significantly (*Figure 3A*). The cell cycle was analyzed by determining the nucleus (N) and kinetoplast (K) configuration and showed an accumulation of 1K1N cells with the

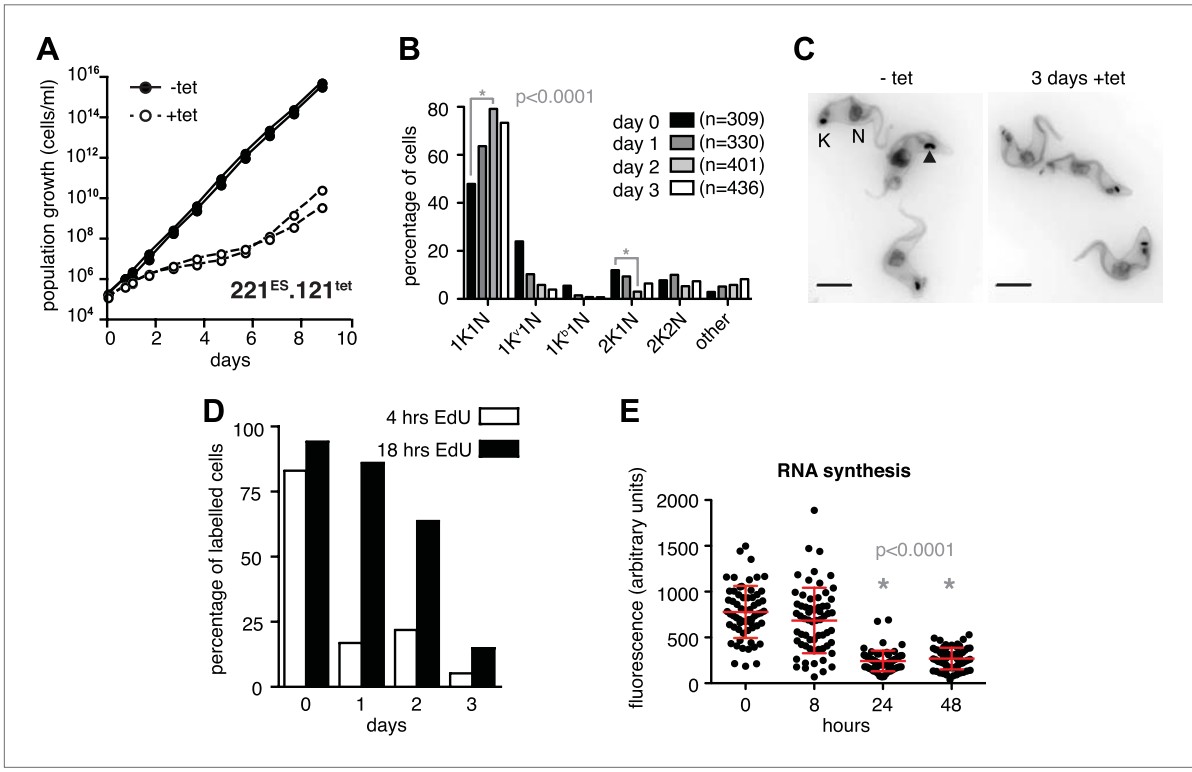

**Figure 3**. Expression site attenuation causes G1 retardation and transcriptional shut-down. (**A**) Cumulative growth curve after induction of ectopic *VSG121* expression. Two independent clones were analyzed for 9 days in the absence (−tet) or presence (+tet) of 1 µg/ml tetracycline. (**B**) Cell cycle analysis. 221ES.121tet cells were induced for *VSG121* expression and fixed at the time points indicated. The configurations of the mitochondrial genome (kinetoplast, K) and the nucleus (N) were visualized by light microscopy following staining with DAPI. The configuration of K and N was subdivided into 1K1N for cells having single copies of each organelle, 1Kv1N for those with an early dividing kinetoplast and 1Kb1N with a late dividing kinetoplast. Cells in G2/M phase reveal 2K1N configuration and post-mitotic trypanosomes are marked by 2K2N. Abnormal configurations are scored as 'other'. For each time point, n >300 cells were analyzed. Significance was determined using Fisher's exact test. (**C**) Cell cycle retardation has no effect on cell morphology. Trypanosomes before (−tet) and 3 days after *VSG121* induction (+tet) were stained with DAPI and the cell surface labeled using AMCA-*sulfo*-NHS. The arrowhead points to a 1Kv1N cell. Scale bar: 5 µm. (**D**) Expression site attenuation leads to reduced growth, but not to cell cycle arrest. Incorporation of EdU into 221ES.121tet cells induced for *VSG121* expression. The trypanosomes were induced for ectopic *VSG121* expression and at 0, 1, 2, and 3 days, they were incubated with 50 µM EdU for both 4 and 18 hr. After chemical fixation the cells were co-stained with DAPI and analyzed by light microscopy (n >100 each). (**E**) Quantification of RNA synthesis rate after ectopic *VSG121* expression. After 0, 8, 24, and 48 hr of tetracycline induction, cells were incubated in the presence of 0.5 mM BrUTP for 15 min at 33°C, subsequently chemically fixed and incubated with a monoclonal mouse anti-BrdU and an Alexa488-coupled goat anti-mouse antibody. The intensities of 3D images (100 images; z-step 100 nm) were measured in summed slice projections using the ImageJ software and are expressed as arbitrary units. To exclude variations in signal intensities due to different positions within the cell cycle, only G1 cells were analyzed. After 8 hr of tet induction, no altered fluorescence intensity was observed. After both 24 and 48 hr, fluorescence intensity was significantly decreased to 30% (p<0.0001; unpaired *t* test). For each time point more than 60 cells were analyzed. Red lines and error bars show mean ± SD.

The following figure supplements are available for figure 3:

**Figure supplement 1**. Growth retardation is accompanied by a reduced transcriptional rate of Pol I and Pol II.

kinetoplasts having a pre-replication morphology and a reduction of cells containing a dividing kineto-plast (*Figure 3B,C*). These observations are consistent with the slowed growth resulting from a G1 phase prolongation. To test this model, the percentage of cells in S-phase was determined at different times after induction of *VSG121* expression. Cells were incubated with the nucleoside-analogue EdU for either 4 or 18 hr (*Figure 3D*). In the control (0 hr), 80% of trypanosomes had entered nuclear S-phase within the 4-hr labeling window. In contrast, after 24 hr of *VSG121* expression only 17% incorporated EdU into nuclear DNA within an equivalent 4-hr window. However, when the parasites were incubated with EdU for 18 hr, 86% had entered S-phase. Thus, the cells were clearly not arrested in the cell cycle, but showed a specific, at least fivefold expansion of the G1-phase from 3 to 15 hr. Remarkably, all trypanosomes remained fully motile and no morphological abnormality or cell death was observed. The marked G1-phase prolongation obviously required a reduction in metabolic flux and anabolic pathways, including protein synthesis, which in turn might be reflected in decreased overall RNA synthesis. Therefore, we measured total RNA synthesis by labeling nascent transcripts with BrUTP and detection with an anti-BrdU antibody. After 8 hr of tetracycline induction, when *VSG221* transcripts had already decreased to 20%, the general transcription rate was not impaired at all. Only after 24 hr was the global transcription reduced to 30% (p<0.0001; unpaired *t* test) (*Figure 3E*). This formally proves that ES attenuation precedes growth retardation and general transcriptional diminution. To test if RNA polymerase I and II transcription was both affected, we inserted luciferase reporter transgenes either downstream of an rDNA promoter or into the *tubulin*-locus (*Figure 3—figure supplement 1*). In both the cases, luciferase activity was down to 30–50% within 2 days. Thus, all tested regions of the genome were slowly silenced. This also explains the gradual decrease in the amount of ectopically expressed *VSG121* mRNA, which started after 24 hr of tetracycline induction (*Figure 1B*) that is at a time when the active *VSG221* ES was already attenuated.

## Expression site attenuation requires histone methylation

The histone methyltransferase DOT1B is involved in the silencing of the active ES during in situ switching (*Figueiredo et al., 2008*). To test if DOT1B was also required for the attenuation of the active ES upon ectopic *VSG* expression, we inserted the tetracycline-inducible *VSG121* gene into *DOT1B*-deleted trypanosomes (Δdot1b.221^ES.121^tet) (*Janzen et al., 2006*). Following induction of *VSG121* expression, the decrease of the ES-resident VSG221 protein occurred with the same kinetics as in the parental cells (*Figure 4A*). However, unlike in the parental cell line the protein levels did not increase again. Furthermore, the cells did not display a growth phenotype (*Figure 4B*). Therefore, we measured ES activity just upstream of the *VSG221* gene by inserting a luciferase reporter transgene into both Δdot1b and parental cell lines (*Figure 4C*). As expected, the luciferase activity decreased rapidly in the parental cells (*Figure 4C*, right panel). However, in the *DOT1B*-knockout trypanosomes luciferase activity was only marginally and transiently reduced at day 2 (*Figure 4C*, left panel). Thus, in *DOT1B*-deleted cells, the ES was not attenuated, except for the most telomere-proximal part, where the *VSG221* gene resides. This suggests that silencing of a *VSG* gene can be uncoupled from ES silencing. While ES attenuation seems to require DOT1B-dependent histone tri-methylation, *VSG* silencing does not. This is remarkable as it suggests a mechanism that specifically represses the telomeric *VSG* gene but not any other part of the ES (*Figure 4A*). Furthermore, these results formally prove that the ES-resident *VSG* can be complemented in trans by a heterologous *VSG* gene. We conclude that the activation of a new *VSG* ES initiates an in situ switch, and that high-level expression of a second VSG is sufficient to attenuate and eventually silence the formerly active ES. This feedback loop assures monoallelic expression of the *VSG*.

## Expression site attenuation induces full developmental competence

There is a period in the life cycle of trypanosomes, when allelic exclusion of *VSG* is lifted, namely during differentiation to the quiescent stumpy stage. This developmental transition occurs in the mammalian bloodstream and involves ES-attenuation and growth arrest. The VSG-induced ES attenuation described here does not cause a bona fide cell cycle arrest and the parasites do not display the typical morphology of stumpy trypanosomes. Nevertheless, we hypothesized that the ES-attenuated parasites could have adopted stumpy-like characteristics and actually may represent the enigmatic intermediate stage, which is not yet committed to stumpy differentiation. Therefore, we tested for expression of the cell surface transporter *protein associated with differentiation 1* (PAD1), which is a molecular marker for stumpy cells and is not expressed in the dividing slender stage (*Dean et al., 2009*). In

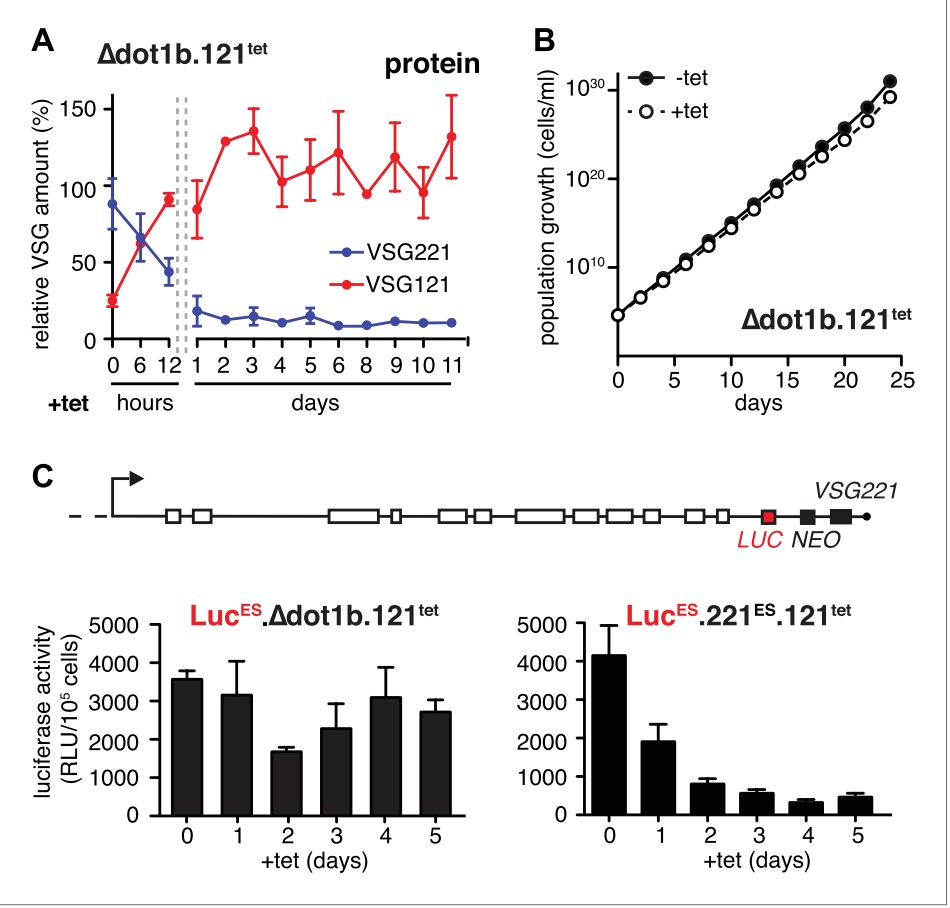

**Figure 4**. Expression site attenuation requires the histone methyltransferase DOT1B. (**A**) Quantification of VSG protein levels during the course of tetracycline-induced *VSG121* expression in *DOT1B*-depleted cells. Quantification was done by normalization to the paraflagellar rod protein (PFR), using the Licor Odyssey system. The values are percentages ± SD for two independent clones normalized to the parental *Δdot1b* or VSG121 wild-type cells. (**B**) Growth curve after induction of ectopic *VSG121* expression in a *Δdot1b* cell line. Two independent clones were analyzed for 25 days in the absence (−tet) or presence (+tet) of 1 μg/ml tetracycline. (**C**) A luciferase reporter gene (*LUC*) was inserted into the active *VSG221* ES (upper panel) of Δdot1b.121[tet] and 221[ES].121[tet] cells. *VSG121* expression was induced in the resulting Luc[ES]Δdot1b.121[tet] and Luc[ES].221[ES].121[tet] cells and luciferase activity of three (Luc[ES]Δdot1b.121[tet]) or two (Luc[ES].221[ES].121[tet]) independent clones was measured at the time points indicated and expressed as relative light units (RLU) ± SD.

addition to a PAD1-specific antibody, we used a transgene, *GFP:PAD_{utr}*, encoding a GFP with a nuclear localization signal and a *PAD1* 3′UTR (J Sunter, A Schwede and M Carrington, unpublished), which confers the stage specificity of PAD1 (**MacGregor and Matthews, 2012**). This reporter allowed for live cell analysis of PAD1-expression by flow cytometry (**Figure 5—figure supplement 1**). Prior to induction of the ectopic *VSG*, the cells showed no PAD1-expression and no GFP signal. After 48 hr of *VSG121* overexpression, however, the PAD1 protein was clearly detectable on the surface of trypanosomes in the G1-phase of the cell cycle (**Figure 5A,B**). Likewise, the GFP:PAD_{utr} reporter was visible in most G1 cells, but was also expressed in a minor portion of trypanosomes in other cell cycle phases (**Figure 5C**). This is in agreement with the observation that the *PAD1* mRNA is detectable prior to the protein (**MacGregor et al., 2012**). We conclude that in the absence of the quorum-sensing factor SIF and at 37°C, the *VSG* overexpression-induced ES attenuation triggers G1-phase prolongation and concomitant expression of a bona fide stumpy marker in monomorphic slender stage trypanosomes. Notably, PAD1 expression was induced despite the global transcriptional attenuation that started at the same time. This again is reminiscent of the situation in stumpy parasites, where PAD1 expression is induced while general transcription decreases (**Amiguet-Vercher et al., 2004**).

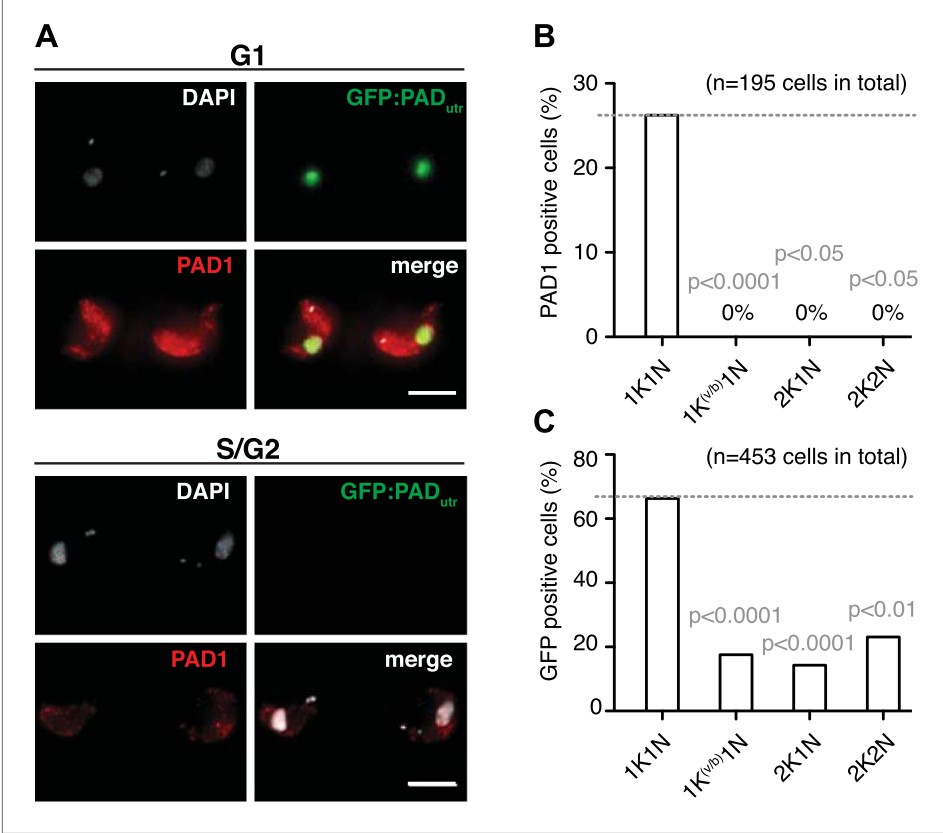

**Figure 5**. Expression site attenuation leads to PAD1 surface expression. (**A**) GFP:PAD$_{utr}$ and endogenous PAD1 are both expressed following ES attenuation. Maximum intensity projections of three-channel, 3D images (100 images, z-step: 200 nm) of GFP:PAD$_{utr}$.221$^{ES}$.121$^{tet}$ cells induced for *VSG121*. After 48 hr, the cells were chemically fixed and incubated with an anti-PAD1 antibody (red) and counterstained with DAPI (grey). The GFP:PAD$_{utr}$ reporter is expressed in the nucleus (green). The upper panel shows an example of G1 cells co-expressing the native PAD1 (red) and the reporter GFP:PAD$_{utr}$ (green). S- and G2-phase cells did not show PAD1 expression (lower panel). Scale bar: 10 μm. (**B** and **C**) Cell cycle distribution of trypanosomes expressing the native PAD1 protein (**B**) and the GFP:PAD$_{utr}$ reporter (**C**). Both the GFP-reporter and the PAD1 protein are expressed specifically in G1 cells. PAD1 expression is limited to G1. GFP:PAD1$_{utr}$ is visible in more G1 cells and also in other cell cycle stages. Values show percentages of positive cells of the indicated cell cycle stages and p-values were calculated using the Fisher's exact test.

The following figure supplements are available for figure 5:

**Figure supplement 1**. Flow cytometry analysis of GFP:PAD$_{utr}$ expression in live cells.

---

Next, we asked if PAD1 was fully functional as a transporter on the cell surface of slender trypanosomes. If so, the parasites should have become sensitive to the differentiation trigger *cis*-aconitate, the chemical cue that triggers progression from the stumpy to the procyclic insect stage. This developmental step is accompanied by loss of the VSG coat and expression of the surface protein EP. We induced *VSG121* expression for 48 hr and then incubated the trypanosomes in the absence of tetracycline with 6 mM *cis*-aconitate. Immunofluorescence analysis revealed EP1 surface expression as early as 6 hr after addition of *cis*-aconitate (*Figure 6—figure supplement 1*). Thus, the mechanism of cell surface access block that prevents EP1 from being routed to the plasma membrane of proliferating bloodstream parasites had been deactivated, just as in stumpy cells (*Engstler and Boshart, 2004*). To quantify the EP1 surface expression, samples were collected 0, 6, and 24 hr after addition of *cis*-aconitate and analyzed by flow cytometry. In the control cells, no fluorescent EP1 signal was detectable (*Figure 6A,B*). In contrast, VSG121-expressors revealed strong EP1 cell surface expression after only 6 hr of *cis*-aconitate treatment. After 24 hr, 70(±6)% of cells were EP1-positive (*Figure 6A,B*). Furthermore, the trypanosomes also altered their cell architecture, adopting a stumpy-like morphology after 6 hr of *cis*-aconitate

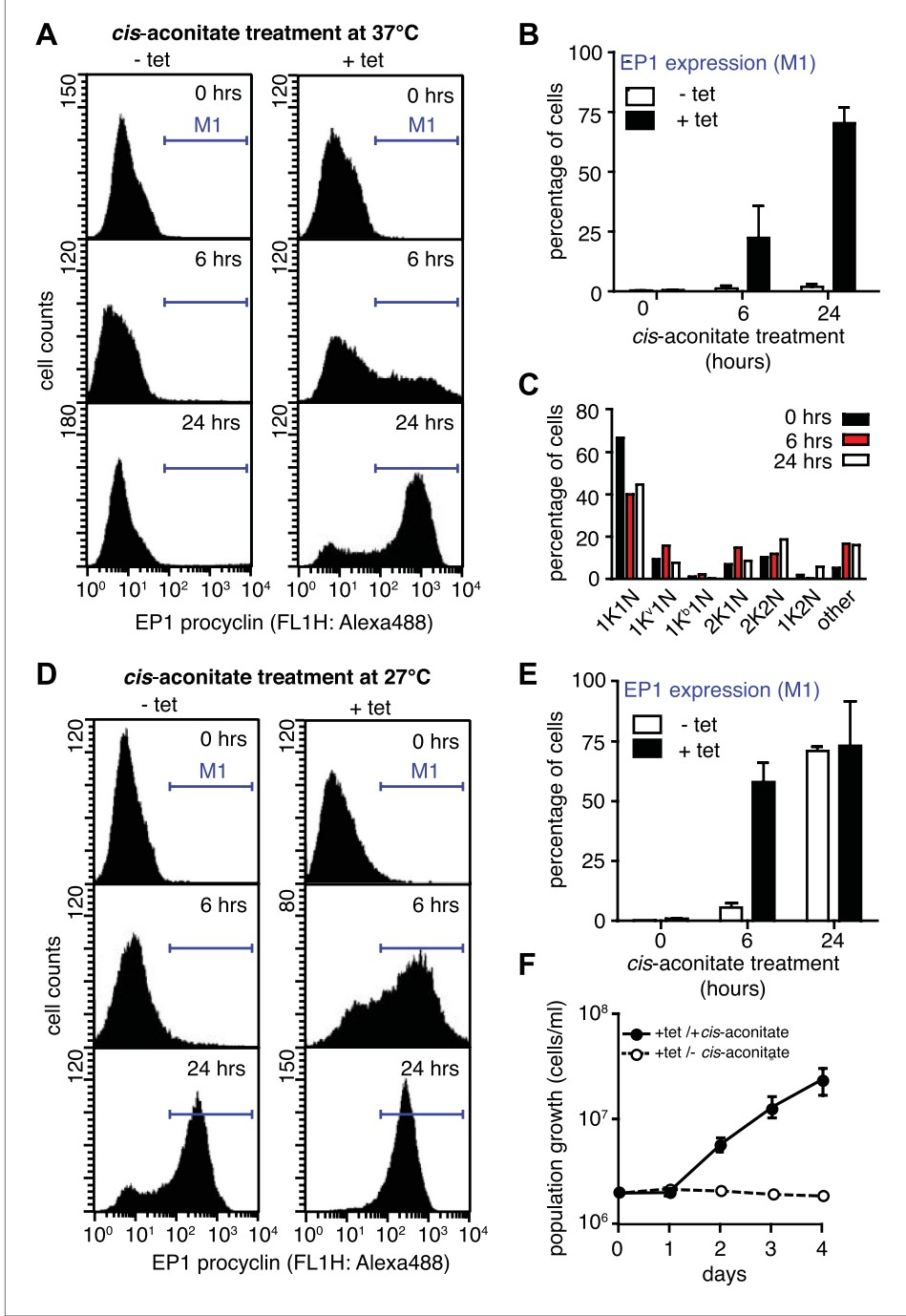

**Figure 6**. G1-retardation renders trypanosomes sensitive to the differentiation trigger *cis*-aconitate and provides full developmental competence at 37°C. The response of trypanosomes to *cis*-aconitate was determined 48 hr after induction of ectopic *VSG121* expression. (**A**) Flow cytometry of EP1 surface expression at 37°C. The induced cell line 221[ES].121[tet] was treated with *cis*-aconitate at 37°C in bloodstream form medium for 0, 6, and 24 hr (+tet). Non-induced cells (−tet) were used as a control. (**B**) Percentage of cells expressing EP1 at the surface in (**A**). Values represent mean ± SD of two independent clones. (**C**) Cell cycle analysis after treatment with *cis*-aconitate for various lengths of time. At least 100 cells were analyzed for each time point. For details see legend to *Figure 2*. (**D**) Flow cytometry analysis of EP1 surface expression at 27°C. The experiments were conducted as described in (**A**) except that cells were grown in differentiation medium DTM at 27°C. (**E**) Percentage of cells expressing EP1 at the surface in (**D**). Values represent mean ± SD of two independent clones. As expected for differentiation at 27°C, after 24 hr of *cis*-aconitate treatment, both populations were expressing EP1. This is due to cold-shock induction of

*Figure 6. Continued on next page*

*Figure 6. Continued*

EP1 in the presence of the dominant differentiation trigger. Values are mean ± SD of two independent clones. (**F**) *Cis*-aconitate treatment releases growth attenuation. After induction for ectopic *VSG121* expression for 48 hr, 221$^{ES}$.121$^{tet}$ cells were transferred to differentiation medium at 27°C in the presence or absence of *cis*-aconitate and kept at a density of $2 \times 10^6$ cells/ml. Cumulative cell numbers per ml are shown as mean ± SD of two independent clones.

The following figure supplements are available for figure 6:

**Figure supplement 1**. Immunofluorescence detection of the insect stage surface protein EP1.

treatment. After 24 hr, the posterior part of the cells had elongated and the kinetoplast had repositioned and segregated (*Figure 6C*, *Figure 6—figure supplement 1*). These morphological changes are hallmarks of the differentiation to the procyclic insect stage.

It is important to note that the above experiments were conducted at 37°C to exclude the possibility that EP1 expression was induced by cold-shock (*Engstler and Boshart, 2004*). As growth of procyclic trypanosomes in fact requires temperatures of 27°C or below and a specific culture medium, we repeated the experiment at 27°C in the differentiation medium DTM (*Overath et al., 1986*). Following tetracycline removal, ES-attenuated trypanosomes were incubated with *cis*-aconitate for 6 and 24 hr, and then analyzed for EP1 expression using flow cytometry. After 6 hr, 58(±8)% of cells revealed strong EP1 fluorescence on the cell surface, while only 5(±2)% of the control population were EP1-positive (*Figure 6D,E*). Interestingly, the ES-attenuated trypanosomes accelerated the cell cycle after 24 hr and started growing as procyclic trypanosomes (*Figure 6F*). This delayed growth response is a characteristic of the differentiation of stumpy trypanosomes to the procyclic insect stage (*van Deursen et al., 2001*) and probably marks the terminal shutdown of the attenuated ES. We conclude that the VSG-induced ES attenuation triggers the cell biological reprogramming of the parasite and causes the concomitant gain of full developmental competence.

## Signaling of expression site attenuation involves three ESAGs

We initiated the shutdown of the active ES by providing wild-type levels of a second VSG. While the overall amount of VSG did not become limiting at any time during our experiments, transcription of *ESAG*s was obviously affected early on (*Figure 2B*). Thus, we surmised that the loss of ESAG homeostasis caused growth retardation and developmental reprogramming of the parasite. As shown above, the reduction of ES activity is gradual, affecting the telomere-proximal *ESAG1* and *2* first. Therefore, we initially examined if loss of these two ESAGs resulted in G1-prolongation and/or PAD1-induction. The GFP:PAD$_{utr}$ cell line was first transfected with an tetracycline-inducible RNAi-construct targeting *ESAG1*. Growth was significantly but transiently impaired at days 2 and 3 of RNAi (*Figure 7A*). The GFP:PAD$_{utr}$ expression was monitored by flow cytometry (*Figure 7B*). Whereas no GFP was detected in the first 24 hr, there was an accumulation of GFP-positive cells after 48 hr and a peak of GFP expression at day 3. Then the GFP-signal decreased again and was close to pre-induction at day 5, when normal growth had resumed.

Next, we knocked down *ESAG2* in GFP:PAD$_{utr}$ cells. No growth defect was observed within the first 2 days of RNAi, but the fraction of GFP-positive cells increased twofold in the first 24 hr. After 3 days, GFP was detected in 75% of all cells. At this time, cell growth stopped and the parasites started to die (*Figure 7C,D*). It should be mentioned that *ESAG1*-RNAi reduced transcript levels to only 30%, while *ESAG2*-RNAi was more effective and reduced the mRNA to less than 5%, which may explain the lethal phenotype.

To examine the possibility that down-regulation of any ESAG induces PAD1 expression, we targeted other ESAGs of the *VSG221* ES. In addition to ESAG1 and 2, only ablation of ESAG8 caused PAD1 induction, albeit in a less pronounced manner. The distinctive growth phenotypes observed after down-regulation of ESAGs 3, 6, 7, and 12 were not accompanied by PAD1 expression at any time (*Figure 7—figure supplement 1*). We also depleted ESAG5 by RNAi and no growth phenotype was observed. These results strongly suggest that G1-retardation and PAD1 expression is caused by the depletion of three ESAGs. Thus, loss of ESAG homeostasis is an immediate early signal for developmental stage transition.

## Expression site attenuation and growth retardation are fully reversible processes

Although ES attenuation provides the cells with full differentiation competence, it does not lead to an irreversible developmental commitment, as is the case in stumpy cells. Starting at day 5 of *VSG121*

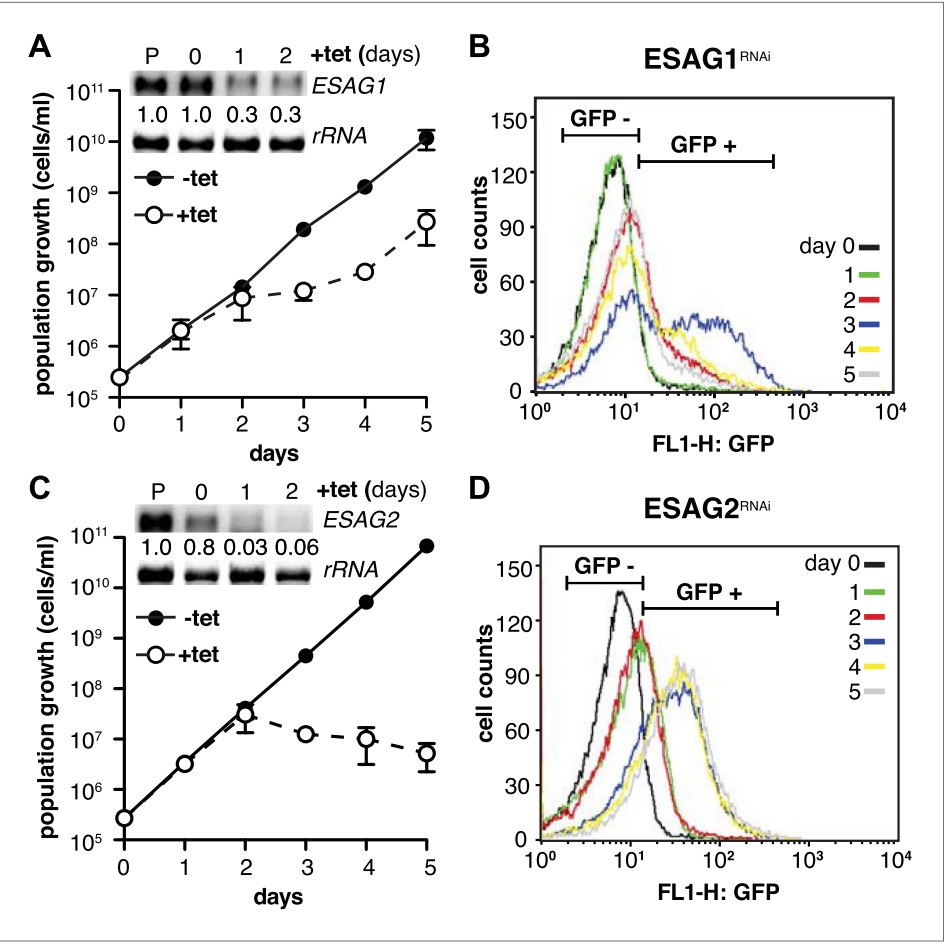

**Figure 7**. Depletion of *ESAG 1* or *2* by RNAi causes distinct growth defects and induction of PAD1 expression. (**A** and **C**) Growth of GFP:PAD$_{utr}$ cells induced for RNAi against either (**A**) *ESAG1* or (**C**) *ESAG2* is presented as the cumulative mean cell number ± SD for two (*ESAG1*) and three (*ESAG2*) independent clones. Upper panels in (**A**) and (**C**) show Northern blots specific for *ESAG1* and *2*, respectively, normalized to 18S rRNA. (**B** and **D**) Flow cytometric analysis of GFP:PAD$_{utr}$ expression in a time course of (**B**) *ESAG1*-RNAi and (**D**) *ESAG2*-RNAi.
The following figure supplements are available for figure 7:

**Figure supplement 1**. Down-regulation of a subset of three *ESAG*s induces GFP:PAD$_{utr}$.

induction, the attenuation of the *VSG221* ES was gradually lifted. Interestingly, the kinetics of ES re-activation was also directional, starting at the promoter and proceeding towards the telomere. Thus, the linearity of ES attenuation was reversed (*Figure 2C*, left panel).

VSG221 mRNA slowly increased to pre-induction levels between days 5 and 8 (*Figure 1B*). At day 8, the VSG221 surface coat had returned and VSG121 was back to pre-induction levels (*Figure 1B*). With onset of *VSG* ES reactivation, the G1-retardation was lifted, cell cycle progression was accelerated, and at day 10, the population doubling times were back to normal (6 hr) (*Figure 3A*). This exact timing was reproducible in many independent experiments with subcloned populations. We have analyzed 20 clonal populations and all clones re-activated the *VSG221* ES; no indication of ES switching was found. We cultivated the trypanosomes for an additional 10 days in the absence of tetracycline and then re-induced expression of *VSG121*. More than 20% of the population was responsive to re-induction and showed strong VSG121 expression, pointing against destructive mutations in the inducible expression system. As an additional control, we have expressed the *VSG121* gene using a constitutive T7 promoter. All resulting clonal populations revealed reduced levels of ectopic VSG121 expression, comparable to inducible VSG121-expressers after ES-re-activation (after 5 days). Importantly, in those cell lines the tetracycline system was still fully functional, as shown by tetracycline induction of an unrelated transgene

(data not shown). These observations strongly argue against selection of revertants that prevent *VSG121* expression by interfering with the T7-system.

Thus, ES attenuation opens a time-window in which the trypanosomes linger in an intermediate state and can progress in two directions, either by differentiating to the insect stage or by returning to proliferation in the mammalian host. This questions the common view that trypanosome development is inevitably unidirectional as in the case of the SIF-induced pathway (*MacGregor et al., 2011*).

## Discussion

The unique feature of the mammalian infective form of African trypanosomes is the plasma membrane with a dense VSG coat. The parasite has evolved an efficient system of antigenic variation that relies on monoallelic expression of *VSG* genes (*Borst, 2002*). Though the system is optimized for a mono-typic coat, two VSGs have to be handled on the same cell surface during antigen switching. Although nothing is known about the length of this period, the different underlying molecular mechanisms suggest variable time scales. Telomere exchange or gene conversion are probably rather fast processes as loss of *VSG* mRNA by RNAi results in rapid cell cycle arrest (*Sheader et al., 2005*). A similar scenario applies if the active ES would be silenced before a new one is activated. In contrast, time would not necessarily be limiting if an in situ switch would involve activation of a new ES before silencing of the old one.

In this study, we simulated the activation of a new ES by high-level, inducible expression of a second heterologous *VSG*. The induction of *VSG121* expression resulted in the rapid decrease of *VSG221* mRNA and protein. The ectopic VSG121 dominated the surface coat for several days. The *ESAG*s were also down regulated, showing that induction of a second *VSG* gene attenuates the entire active ES. The insertion of reporter genes showed that this ES attenuation starts at the telomere and progresses towards the promoter region. Our data further indicate that the initial spike in the production of ectopic VSG acts as an immediate early signal, which is relayed to an epigenetic ES silencing machinery that requires the histone methyltransferase DOT1B. This model further explains the expression levels in stable VSG double-expressors that have a second *VSG* inserted into the active ES, just upstream of the endogenous copy (*Muñoz-Jordán et al., 1996*). In these double-expressors, both VSGs are expressed in a 50/50 ratio, not exceeding a total amount of 100% compared to wild-type single expressors (data not shown). The generation of such double-expressors requires antibiotic selection for 7 days after transfection. In that time, the trypanosomes react to the initial overexpression of VSG, which occurs immediately after recombination of the transgene into the active ES, and attenuate the ES. When the VSG expression is adapted, attenuation is hold, not affecting the upstream located *ESAG*s. Thus, high-level expression of a second *VSG* alone is sufficient to initiate attenuation of the active ES. Importantly and unlike the endogenous *VSG*, this second *VSG* does not have to be expressed from an ES and it does not need to be transcribed by RNA polymerase I. Furthermore, *VSG* silencing and shutdown of the active ES are mechanistically distinct events, as in *DOT1B*-knockout trypanosomes, the ES-resident *VSG* is silenced upon *VSG* overexpression, but the *ESAG*s are not. No phenotype is observed after *VSG* overexpression in the *DOT1B*-knockout cells, proving that neither the initial surplus of VSG production nor the non-trypanosomal T7 RNA polymerase system is toxic.

In the mammalian olfactory receptor (OR) system, the expression of an ectopic *OR* transgene is sufficient to mediate a feedback inhibition that prevents the expression of the active endogenous *OR* gene, independent of the receptor signal transduction pathway (*Nguyen et al., 2007*). This is strongly reminiscent of the VSG-mediated attenuation of the trypanosome ES reported here. Despite the enormous evolutionary distance between kinetoplastids and mammals the mechanistic principle appears to be conserved. The trypanosome model obviously offers some opportunities that are not readily available in the OR system, such as following the kinetics of initiation and reversal of gene silencing. Furthermore, in trypanosomes an array of genes is attenuated, whereas in the olfactory system only the *OR* is silenced. Importantly, the OR system fulfills a single function, namely detection of an almost infinite number of odorant molecules. Our work suggests that the control of monoallelic expression of *VSG* links at least three key-processes, namely antigenic variation, host range determination, and parasite development. Our findings clearly support a model in which the *VSG* ES orchestrates events at the crossroads of antigenic variation, cell cycle, and development. The trypanosome cell can operate for significant periods in a dormant state. This ability may become crucial in the course of an in situ switch to a non-functional or impaired VSG or upon activation of a defective ES. Any trypanosome with an

attenuated ES will experience a time window of full developmental competence. Thus, the stumpy life cycle stage with its genuine cell cycle arrest can be bypassed.

While our experimental system clearly initiates *VSG* switching, it is not suited for completion of the process, most likely because it does not provide functional *ESAG*s. Interestingly, the cells react by not fully silencing the ES, but rather attenuating its activity to about 20%, probably by chromatin remodeling after histone methylation. In response to ES attenuation trypanosomes enter a semi-quiescent state with a specifically extended G1-phase. The parasites remain in this slow growth state for about 5 days and then resume normal growth without any sign of reduced fitness. While ES attenuation is triggered by the brief period of true *VSG* overexpression, the entry into the dormant phase is the consequence of ESAG shortage. It is important to note that ES attenuation does not lead to complete ESAG exhaustion, but rather to gradual reduction in expression. Thus, individual ESAGs could reach growth-limiting levels at different concentrations and hence, could induce G1-retardation at different times. In fact, depletion of *ESAG1* mRNA by RNAi leads to transient growth retardation after 48 hr. The cells recover from this phenotype 2 days later, which is in agreement with the existence of *ESAG1* knockout lines (*Carruthers et al., 1996*). Importantly, even depletion of the non-essential and ES-specific *ESAG8* (*Hoek and Cross, 2001*) becomes growth limiting for a transient period. In contrast, *ESAG2*-RNAi is irreversibly lethal and therefore, a complete ES shutdown without functional rescue from non-ES copies or a new ES would be detrimental. Although essential, the down-regulation of *ESAG2* to the level prevailing in ES-attenuated cells (20%) is tolerated. However, the function of these ESAGs remains elusive. ESAG1 and 2 are N-glycosylated, membrane-associated proteins localized in the flagellar pocket. It is suggested that both could function in the endosome pathway (*Nolan et al., 1999*; *Pays et al., 2001*). Thus, ESAG1 and 2 could act in concert, which would explain the in parts redundant RNAi phenotypes. In contrast, ESAG8 is found in the nucleolus and the cytoplasm, and interacts with a Puf family member (*Hoek et al., 2000*; *Hoek et al., 2002*). This suggests that ESAG8 plays a role in mRNA regulation and might be linked to the differentiation pathway.

Thus, the concept of transient ES attenuation rather than immediate and complete ES silencing may be of some physiological relevance; it could allow probing for ES performance, for example ESAG functionality, VSG switching orders, quality of novel mosaic VSGs, or newly assembled expression sites. If both VSGs are compatible, and produce a mixed and dense surface coat and the *ESAG*s of the new ES are fully functional, the old expression site is rapidly and completely silenced with no effects on cell cycle progression. However, if any component of the new ES fails or is not sufficiently functional, the cells can adjust growth, in the extreme, to very basal levels. This way the parasites may either gain time to adapt to the new conditions or reverse the attenuation and re-activate the old ES. We suggest that the latter case resembles an unsuccessful switching event, which would escape experimental detection. In fact, we can only observe successful *VSG* switching events, thereby neglecting the surprising capacity of trypanosomes to reverse switching directions.

This flexibility could become particularly productive in varying host environments. Provided a sufficiently high ES activation frequency exists, the parasites could effectively probe for host serum-compatible ESAGs (*Bitter et al., 1998*; *Chaves et al., 1999*; *Pays et al., 2001*). Hence, ES switching plasticity could determine the host range. One extreme example for this is *Trypanosoma b. rhodesiense,* which is human-infective and thought to be a host-range variant of *T. b. brucei*. The parasites are specifically adapted to human serum by expression of the serum-resistance antigen (SRA) from one particular *VSG* ES in most isolates. The structure of the *SRA*-containing ES is unusual as it is missing several *ESAG*s present in all other ESs. During an infection of a human, an in situ switch to a *VSG* in another ES would cause loss of *SRA* expression, with lethal consequences. Thus, *T. b. rhodesiense* must not switch off the *SRA*-containing ES, thus making the ESAG-dependent differentiation pathway obsolete. This may be the reason why the *T. b. rhodesiense* ES has lost *ESAG1*, *2* and *8*.

The fact that ES attenuation causes growth retardation could also become mechanistically relevant during another phase of the trypanosome life cycle, namely upon differentiation to the G1/G0-arrested stumpy stage. The developmental transition from the proliferating slender to the quiescent stumpy bloodstream stage is initiated by quorum sensing of the trypanosome-derived compound SIF (*Reuner et al., 1997*). The monomorphic cells used in this study are unresponsive to SIF and hence, should have lost the capacity to differentiate to the tsetse-infective stage (*Vassella et al., 1997*). In our experiments, however, VSG-induced ES attenuation renders monomorphic trypanosomes

fully competent for development. After 2 days of *VSG* overexpression the stumpy stage marker PAD1 was strongly up-regulated, specifically in G1-phase cells, and despite global transcriptional silencing being initiated at the same time. This indicates a highly specific event, which, in the absence of SIF, is orchestrated solely by the state of the *VSG* expression site. We are currently investigating this alternative pathway in a pleomorphic strain. The induced expression of a second *VSG* indeed leads to the formation of stumpy forms (data not shown). While VSG itself is not involved, as it is not limiting in our system, three ESAGs signal ES attenuation to the differentiation pathway. Depletion of *ESAG*s *1*, *2* and *8* mRNA by RNAi resulted in induction of the PAD1 reporter GFP:PAD$_{utr}$. Down-regulation of any of the other *ESAG*s did not cause GFP:PAD$_{utr}$ expression. This underlines the specificity of the pathway and suggests that ES activity controls the induction of transient developmental competence. Remarkably, by embedding the endogenous PAD1 protein into the VSG coat, the ES attenuated trypanosomes became fully responsive to the developmental trigger *cis*-aconitate. They initiated transformation to the procyclic insect stage with the same kinetics as stumpy parasites.

There is, however, one principal feature that distinguishes SIF-mediated stumpy formation from ES-regulated cell dormancy. While stumpy parasites are terminally differentiated and can only survive if taken up by an insect, the onset of differentiation is fully reversible in the ES-attenuated trypanosomes. We postulate that these parasites linger in an intermediate state with open developmental choices (*Figure 8*). They have the capacity to complete ES switching by adapting to the new ES or, alternatively, they can re-activate the old, attenuated ES. In both of these cases the parasites resume growth as mammalian bloodstream form cells. If taken up by the tsetse fly, however, they would immediately transform to the procyclic insect stage.

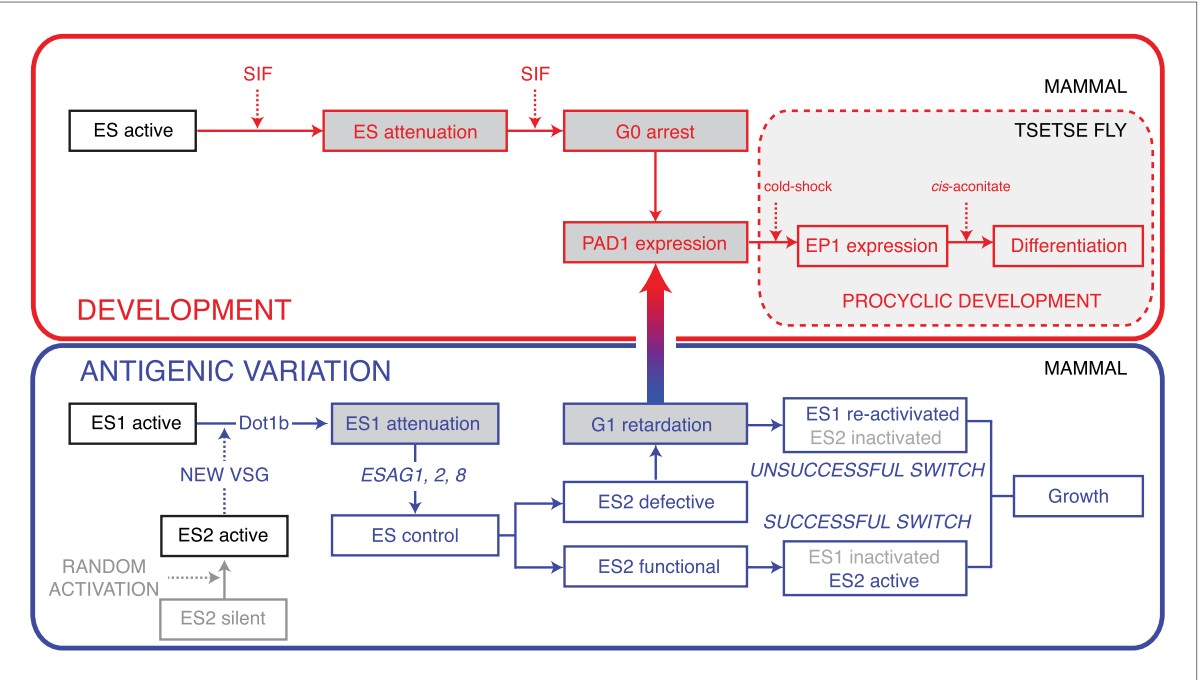

**Figure 8**. Model of the dual roles of expression site attenuation for antigenic variation and development of *T. brucei*. Activation of a new ES leads to attenuation of the previously active one in a DOT1B-dependent manner. The attenuation and subsequent growth retardation allows ES quality control via ESAG1, 2 and 8. If the newly activated ES is functional, the old one is rapidly silenced, resulting in a *successful* VSG switch. If, however, a non-functional or non-compatible ES is activated, the cells react with a specific prolongation of the G1-cell cycle phase. This dormancy can last for up to 5 days and is fully reversible. Within this period the trypanosomes become competent for developmental stage transition, which involves expression of the stumpy marker PAD1. The stumpy stage, however, is irreversibly arrested in G0 and therefore committed to a life in the tsetse fly. In contrast, ES attenuation causes a reversible G1 dormancy, which allows either re-activation of the attenuated ES (*unsuccessful switch*) or survival by development in the insect vector. Thus, ES attenuation operates at the crossroads of antigenic variation and development.

This high flexibility might become biologically relevant especially during early infections, when certain ESAGs or VSGs are not compatible with the new host environment: cells either adapt or regain fly-infectivity. Thus, the state of the expression site must not be regarded as a simple on-off switch, but rather represents a fine-tuned module at the unexpected crossroads of antigenic variation and development.

## Materials and methods

### Trypanosome culturing and transfection

All cell lines generated in this study are based on *T. brucei* 427 MITat 1.2 13-90 bloodstream forms (221[ES]) (*Wirtz et al., 1999*) and were cultivated in HMI-9 medium with 10% fetal bovine serum (Sigma-Aldrich, St. Louis, MO) at 37°C and 5% $CO_2$. For T7-polymerase and tetracycline repressor maintenance, cells were selected with 5 µg/ml hygromycin and 2.5 µg/ml G418. For transfections, $3 \times 10^7$ parasites were electroporated with 10 µg of linearized plasmid DNA using the AMAXA Nucleofector II (Lonza, Switzerland). Details of expression constructs and cell lines are provided in *Supplementary file 1*.

### Plasmid construction and generation of cell lines

The VSG121 open reading frame with the full 3′UTR was amplified from MITat1.6 wild-type genomic DNA with the primers VSG121_fw and VSG121_rev, subcloned into a pBluescriptSK, mobilized with HindIII and cloned into the pLew82v4 (24,009; Addgene plasmid). The resulting plasmid pRS.121 was NotI-linearized and transfected into the bloodstream form 221[ES], giving rise to the 221[ES].121[tet] trans-formants. pkD.GFP was generated by cloning a 1.6 kb SpeI/EcoRI fragment containing a *GFP* ORF flanked by VSG221 5′ and 3′UTRs from p3845 (kindly provided by A Schwede and M Carrington, Cambridge, UK) into the VSG221 ES targeting vector pkD (*Muñoz-Jordán et al., 1996*). Cell lines GFP[ESpro]-221[ES].121[tet] and GFP[EStel]-221[ES].121[tet] were generated by inserting p3845 and pkD.GFP, respectively, into 221[ES].121[tet]. For the Luc[ES] cell lines, the luciferase gene was mobilized with HindIII from pLew82v4, overhangs were filled in and the resulting product was ligated to an EcoRI digested and blunted pkD vector. The resulting plasmid pkD.Luc was linearized with AvrII and KpnI and transfected into 221[ES].121[tet] or Δdot1b.121[tet]. pRib.Luc was generated by cloning the luciferase gene between the HindIII and BamHI sites of pTSA-Rib (*Xong et al., 1998*), in which the tubulin 3′UTR had been replaced by the PARP 3′ region and the hygromycin resistance by puromycin resistance. The resulting plasmid was SphI linearized and transfected into 221[ES].121[tet], creating Luc[rPro].221[ES].121[tet]. pTub.Luc was created by replacing a BsaBI/SphI fragment from pHD328 (*Wirtz et al., 1994*) with a BsaBI/SphI fragment from pLew20 (*Wirtz et al., 1999*) and the *ble* gene was replaced by a puromycin resistance gene. The resulting plasmid was linearized with NotI and transfected into 221[ES].121[tet], resulting in Luc[tub].221[ES].121[tet]. For generating the GFP:PAD[utr] reporter cell lines, the plasmid p4231 (J Sunter, A Schwede and M Carrington, unpublished) was transfected into the parental 221[ES] or the 221[ES].121[tet] cell line. For RNAi, the target sequences were amplified from genomic DNA of 221[ES] cells using the following primers: ESAG1_fw (5′-TTGTGTTGATGCATG-3′) and ESAG1_rev (5′-TCGGTCTTG GTTTAG-3′); ESAG2_fw (5′-GAAATAGTGATTGCCG-3′) and ESAG2_rev (5′-CAAACTCAGCTAATGC-3′); ESAG3_fw (5′AAAAA AAAGCTTTCCTTC AAGATGAAG AAGC-3′) and ESAG3_rev (5-AAAAAAGGATCCAAACAAGTCATT CTCCTTGACC-3′); ESAG6/7_fw (5′-AAAAAAAAGCTTGTTTTGGTTTGTGCTGTTGG-3′) and ESAG6/7_rev (5′-AAAAAAGGATCCATACTTTCCGCACCCAAGC-3′); ESAG8_fw (5′-GCACTACGTGATCTGG AAGC-3′) and ESAG8_rev (5′-CATAGAGCACCCTC AAGTGG-3′); ESAG12_fw (5′-AGCGGTGTC AATATTC-3′) and ESAG12_rev (5′-AGGAGGAAGGAGTTTG-3′). The PCR products were subcloned into a pBluescriptSK+ and mobilized with HindIII and SpeI (*ESAG1* and *2*), HindIII and BamHI (*ESAG3*, *ESAG6/7*), or XbaI and XhoI (*ESAG8*) to ligate them into the RNAi plasmid p2T7 (*Shi et al., 2000*). The resulting plasmids were linearized with NotI and transfected into the GFP:PAD[utr].221[ES] cell line.

### EdU and BrUTP labeling

Newly synthesized DNA was labeled with 5-ethynyl-2′-deoxyuridine (EdU) using the Click-iT EdU Alexa Fluor 488 Imaging Kit (Invitrogen, Carlsbad, CA), essentially following the manufacturer's instructions. The cells were incubated with 50 µM EdU in HMI-9 medium, washed in buffer and chemically fixed. After permeabilization with 0.1% Triton X-100, the reaction cocktail was added and the cells were washed twice. BrUTP labeling of nascent transcripts was achieved as described in *Smith et al. (2009)*.

## RNA and protein analyses

Total RNA was isolated from $1 \times 10^8$ trypanosomes using the Qiagen RNeasy Mini Kit (Qiagen, Netherlands) essentially following the manufacturer's instructions. For RNA quantification with fluorescently labeled probes (GFP-probe: GCCGTTCTTCTGCTTGTCGGCCATGATATAGA; VSG121-probe: GGCTGCGGTTACGTAGGTGTCGATGTCGAGATTAAG; VSG221-probe: CAGCGTAAACAACGCACCC TTCGGTTGGTCGTCTAG; Tubulin-probe: ATCAAAGTACACATTGATGCGCTCCAGCTGCAGGTC; 18SrRNA-probe: CAACCAAACAAATCACTCCACCGACCAAAA), 3 µg of total RNA was denatured with glyoxal and blotted with a Minifold Dotblotter (Schleicher & Schuell, Germany) onto nitrocellulose and hybridized overnight at 42°C. *ESAG* transcripts were detected with $^{32}$P-labeled probes and quantified using a Phosphorimager. To precipitate protein, the cell lysates were mixed with ice-cold acetone and centrifuged at 20,000×*g* for 10 min. The sediment was air-dried and resuspended in sample buffer (2% SDS, 10% glycerol, 60 mM Tris–HCl, pH 6.8, 1% β-mercaptoethanol) to yield equivalents of $2 \times 10^5$ cells/µl. For protein quantification, 3 µl was spotted onto nitrocellulose, air-dried, and incubated with specific polyclonal anti-VSG antibodies (rabbit anti-VSG221/C-term, 1:5000; rabbit anti-VSG121, 1:2000) and mouse monoclonal anti-PFR antibody (L13D6, 1:20). For fluorescence detection, an IRDye680-conjugated goat-anti-mouse antibody and an IRDye800-conjugated goat-anti-rabbit antibody (1:10,000; Licor, Lincoln, NE) were used. Protein and RNA were analyzed and quantified using the Licor Odyssey System.

## Luciferase assay

$1 \times 10^6$ to $1 \times 10^7$ trypanosomes were washed in ice-cold PBS and resuspended in 100 µl of cell culture lysis buffer (Promega, Madison, WI). 5 µl of lysate were added to 45 µl of luciferase substrate and the luminescence was immediately measured with a Tecan Infinite M200 plate reader (Tecan, Switzerland).

## Immunofluorescence analysis and flow cytometry

IF for surface detection of procyclin EP1 was done as in *Engstler and Boshart (2004)*. For VSG IF, chemically fixed cells were incubated with a mouse anti-VSG121 (1:100) and a rabbit anti-VSG221 (1:100) antibody, followed by incubation with an Alexa488- and Alexa594-conjugated anti-mouse and anti-rabbit antibody, respectively (1:1000). For PAD1 detection, chemically fixed cells were permeabilized with 0.05% Triton X-100 for 10 min and incubated with a rabbit anti-PAD1 antibody (1:100), followed by incubation with an Alexa488-conjugated anti-rabbit antibody (1:1000). Flow cytometry was performed with a BD Bioscience FACSCalibur Flow Cytometer. Data were analyzed with the BD CellQuest Pro Software. For each sample, 20,000 cells were counted.

## Acknowledgements

This work was supported by the Deutsche Forschungsgemeinschaft (grant EN 305 to ME). We are grateful to M Carrington (University of Cambridge, UK) for helpful discussion and critical reading of the manuscript. We thank A Schwede (University of Cambridge, UK) for the plasmid p3845, J Sunter (University of Oxford, UK) for the plasmid p4231, M Carrington and K Matthews (Edinburgh) for providing antibodies. We thank L Figueiredo (Universidade de Lisboa), N Siegel, and A Hartel (Universität Würzburg) for discussion.

## Additional information

### Funding

| Funder | Grant reference number | Author |
|---|---|---|
| Deutsche Forschungsgemeinschaft (DFG) | EN 305 | Markus Engstler |
| DFG Sonderforschungsbereich (SFB) 630 | SFB 630-B8 | Markus Engstler |

The funder had no role in study design, data collection and interpretation, or the decision to submit the work for publication.

### Author contributions

CB, Conception and design, Acquisition of data, Analysis and interpretation of data, Drafting or revising the article; NGJ, CJJ, Analysis and interpretation of data, Drafting or revising the article;

SMM, Acquisition of data, Analysis and interpretation of data; ME, Conception and design, Analysis and interpretation of data, Drafting or revising the article

## Additional files

**Supplementary file**
• Supplementary file 1. Cell lines and targeting constructs.

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
