## [Decision Letter]

Thank you for sending your work entitled “Expression site attenuation mechanistically links antigenic variation and development in *Trypanosoma brucei*” for consideration at *eLife*. Your article has been favorably evaluated by a Senior editor, a Reviewing editor and 3 reviewers.

The Reviewing editor and the other reviewers discussed their comments before reaching this decision, and the Reviewing editor has assembled the following comments to help you prepare a revised submission.

We found your work very interesting, potentially revealing new insight into a central process of parasite survival in the host – the ability to switch surface antigens and evade the host immune response. This would be a valuable new view on this central process, revealing parallels with permanent monoallelic gene controls amongst gene families in mammals. However, we also agreed that there were potential problems with the work, outlined below, which we would like you to address. The two most major issues are first, whether your experimental strategy really does mimic VSG switching, and the extent to which this might causes an artefact. Second, there is concern related to the suggested trigger for VSG switching: expression of a new VSG. It was also felt that in places conclusions are over-stated or the most current literature not taken into account.

1) Since ectopic T7-based expression of *VSG* is non-physiological and this triggers a global reduction in transcription, it is unclear to what extent certain phenotypes actually reflect physiologically relevant processes as proposed. The authors work hard to convince the reader that the events observed are consistent with processes associated with *VSG* switching and with differentiation of replicative to non-replicative cells. However, these data might also be explained by over-expression of *VSG121* being toxic to the cells, at least in the conditions used. For example, the growth, replication and gene expression effects that are observed might be explained by the cells becoming very sick upon *VSG121* induction. Why this might be is unclear, but one possibility is that very high levels of mRNA for *VSG121*, or perhaps very high levels of two VSGs, overwhelm the transcription and translation apparatus of the cell. If this were the case, it would explain why the cells stop growing and the effects seen. The authors should comment on this.

The observed return to growth after 5 days, associated with re-expression of *VSG221* (in the ES), is of particular concern. Could this represent the selection for 'revertants' that suppress ectopic expression of *VSG121*? The authors note (in second paragraph of the Results section entitled “Expression site attenuation and growth retardation are fully reversible processes”) that 'the trypanosomes became responsive to re-introduction of tetracycline', which might suggest that permanent mutations that prevent *VSG121* expression have not been selected. However, the wording used is rather vague, and it is not at all clear whether this is their meaning. In addition, we are provided with no evidence to support the putative claim that *VSG121* can be re-induced at this stage. The authors must comment on this further: if revertants are being selected for that no longer express *VSG121*, it suggests significant problems with the experimental approach.

A main argument that the authors make to suggest that the *VSG121* expression is an accurate mimic of *VSG* switching and differentiation, is that they see a general decrease in transcription, consistent with a previous report by [2]. In more recent work, using whole-genome gene expression analyses (Kabani et al, 2009, BMC Genomics; Capewell et al, 2013, PLoS ONE), there is little evidence for such global reduction in transcription: though translation decreases in short stumpy cells relative to long slender, only a very small number of genes show reduced levels of mRNA (e.g., 41 in the Capewell study). Thus, the global effects described here upon *VSG121* induction may not reflect the developmental change from long slender to short stumpy cells, but instead toxic effects of *VSG121* overexpression. Thus, the association between *VSG* switching and differentiation may not be warranted. The authors must discuss these data in the context of their model.

It is currently not convincing that the *VSG* expression site is a 'global regulator' (end of the Introduction) or 'global control module' (end of the Discussion) of antigenic variation, proliferation and development. Without studies in a fully differentiation-competent 'pleomorphic' strain, strong conclusions can't be drawn. These sections should be toned down to reflect the uncertainty.

2) The strategy used to induce 'switching' involves expression of second VSG, and the authors suggest that such double VSG expression is the trigger for switching – e.g., in the final paragraph of the Introduction “Thus, the VSG itself mediates a feedback loop that controls ... monoallelic gene expression”; “an in situ switch would involve activation of a new ES before silencing of the old one' (first paragraph of the Discussion); and the authors draw parallel with mammalian OR silencing (third paragraph of the Discussion) ”...expression of an ectopic or transgene is sufficient to mediate a feedback inhibition....”. If this is the case, the authors must explain some anomalies. First, this model predicts that two VSGs cannot be simultaneously expressed, as one will trigger suppression of another. However, double *VSG* expressers have been reported, and it has been shown that such double expressers are stable in vitro and in vivo, and do not impair growth: Baltz et al. (1986), Nature; and Munoz-Jordan et al. (1996), Science. How can these data be explained, in the context of the model that the authors propose? Second, the authors' model appears to suggest that expression of a second VSG begins a progressive attenuation of the ES from the telomere, allowing the cells to 'probe for ES functionality'. However, how can this actually happen, since in the normal context a second VSG can only be expressed in conjunction with the upstream *ESAGs*, since they share a common promoter in the ES? In other words, it seems that the only circumstances in which *ESAG*s are not co-expressed with the *VSG* is when an ES is activated that lacks these genes; if this is the case, it is only in these circumstances that G1 dormancy and reversible differentiation occurs (these are induced by LOSS of some ESAGs, as shown by this work). If a new *VSG* is expressed from an ES with a full complement of *ESAGs*, these phenotypes would not be seen, further suggesting that they are an artefact of ectopic *VSG121* expression from outside an ES.

3) Expression site transcription may control developmental competence via at least three genes co-transcribed with the VSG but it's difficult to see how each gene knockdown acts redundantly to bring about this effect. Could the authors say more about the putative functions of these *ESAGs* relative to those not linked to this phenotype? Is a SIF signal thought to downregulate these ESAGs and in-turn render the cells more competent for differentiation or is *ESAG* downregulation thought to bypasses SIF signalling?

4) The ESAG sensing model states that *ESAGs 1, 2* and to some extent *8* are important while the other *ESAGs* are not. However, *ESAG 5*, present in all known ES was not tested. The data would be strengthened if ESAG5 RNAi was included.

5) The authors’ results quantitating transcripts using hybridisation of dot-blots or Northern blots rather than by qPCR was not completely convincing. In the first Results section the authors say “It is immediately apparent that mRNAs.. decreased faster...” These quantifications should ideally be repeated using qRT-PCR*. ESAG1* and *ESAG2* are members of very large gene families, which contain copies outside of ESs, also transcribed by Pol II. Using qPCR, the authors could design primers specific for the *ESAG* copies present in the *VSG221* ES (and not detect non-ES derived *ESAG*s).

[Editors' note: further clarifications were requested prior to acceptance, as described below.]

1) One point in particular remains unclear, which is an important issue: the authors say that “it is certainly possible to express two *VSG* ES at the same time, albeit not for long and not at the same level” and then that “cells must adapt to the production of the two VSGs in order to survive. They do this by balancing their expression levels – exactly as our model would predict”. This seems contradictory. Would the model not predict attenuation of the downstream *VSG* in the experiments reported by Munoz-Jordan et al. (1996)?

2) In the absence of knowing the functions of *ESAGs 1, 2* and *8*, the evidence that *ESAGs1, 2* and *8* serve as signals for ES attenuation still a bit speculative. I think that the Abstract should read that these “appear to serve” or “could serve” as a signal for ES attenuation.

---

## [Author Response]

*1) Since ectopic T7-based expression of* VSG *is non-physiological and this triggers a global reduction in transcription, it is unclear to what extent certain phenotypes actually reflect physiologically relevant processes as proposed. The authors work hard to convince the reader that the events observed are consistent with processes associated with VSG switching and with differentiation of replicative to non-replicative cells. However, these data might also be explained by over-expression of* VSG121 *being toxic to the cells, at least in the conditions used. For example, the growth, replication and gene expression effects that are observed might be explained by the cells becoming very sick upon* VSG121 *induction. Why this might be is unclear, but one possibility is that very high levels of mRNA for* VSG121*, or perhaps very high levels of two VSGs, overwhelm the transcription and translation apparatus of the cell. If this were the case, it would explain why the cells stop growing and the effects seen. The authors should comment on this*.

A) The overexpression of *VSG* per se is non-toxic. In *DOT1B* knockout cells, tetracycline-induced *VSG121* expression occurs with identical kinetics and to the same level as in wild type cells (Figure 1). Importantly, in these mutant cells, no growth phenotype or any transcriptional shutdown occurs (Figure 4). This means that the surplus *VSG* mRNA or VSG protein does not overwhelm the transcription or translation apparatus of the cells. Thus, the highly specific growth and gene expression phenotype observed in wild type cells is a consequence of expression site attenuation, which does not occur in the *DOT1B* knockout cells.

B) As a control, we have overexpressed a GFP transgene to similarly high levels. This was indeed toxic to the trypanosomes. The cells started to die within 24 hours and revealed grossly abnormal morphologies. Thus, the GFP overexpressors behaved fundamentally different from trypanosomes after induction of ectopic *VSG* expression.

C) The T7-based gene expression system was intentionally preferred over a polymerase I-based expression system to minimize interference with the endogenous transcription machinery.

*The observed return to growth after 5 days, associated with re-expression of* VSG221 *(in the ES), is of particular concern. Could this represent the selection for 'revertants' that suppress ectopic expression of* VSG121*?*

We cannot formally exclude that spontaneous mutations are responsible for the reproducible suppression of ectopic *VSG121* expression. However, this possibility is extremely unlikely as several new experiments suggest that an epigenetic mechanism causes *VSG121* silencing 5 days after induction:

A) We have now induced *VSG121* expression in a population for 72h followed by limiting dilution to 1 cell/ml and cloning on a 96-well plate. A non-induced culture was diluted and cloned in the same way and served as a control. Seven days later, 18 and 14 clones were isolated from the induced and non-induced population, respectively. By protein dot blot we found that *VSG221* expression was fully restored in all clones. It is extremely unlikely that 18 independent mutation events have by chance and at the same time ‘decommissioned’ the T7 expression system. Therefore, we assume that an epigenetic mechanism is responsible for full re-activation of the attenuated ES and silencing of the ectopic *VSG121* copy.

B) We analysed *VSG* expression at single cell resolution using immunofluorescence analysis (Figure 1—figure supplement 1). All cells in the population were cell surface-positive for *VSG121* within 24 hours after tetracycline induction. This excludes that the return to growth was due to the presence of non-inducible cells in the original population.

C) Furthermore, we have expressed the *VSG121* gene using a constitutive T7 promoter, i.e., in the absence of tetracycline. All resulting clonal populations revealed reduced levels of ectopic *VSG121* expression, comparable to inducible *VSG121*-expressors after ES-re-activation (after 5 days). Importantly, in those cell lines the tetracycline system was still fully functional, as shown by tetracycline induction of an unrelated transgene. These observations strongly argue against selection of revertants that prevent *VSG121* expression by interfering with the T7-system.

D) Within 5 days the attenuated ES-linked *VSG221* is gradually de-repressed and we suggest that it is the increasing amount of reappearing *VSG221* that decreases the levels of the ectopic *VSG121*. Once the *VSG221* ES has been fully re-activated, the ribosomal spacer copy of *VSG121* has been silenced to pre-induction levels (Figure 1). However, in the *DOT1B* knockout cells no suppression of the ectopic copy is observed, even after 10 days of induction (Figure 4). This suggests that depletion of *DOT1B* interferes with an epigenetic silencing mechanism. Similar effects were observed when the in situ switching kinetics was monitored in a *DOT1B-*depleted strain (14).

*The authors note (in second paragraph of the Results section entitled “Expression site attenuation and growth retardation are fully reversible processes”) that 'the trypanosomes became responsive to re-introduction of tetracycline', which might suggest that permanent mutations that prevent* VSG121 *expression have not been selected. However, the wording used is rather vague, and it is not at all clear whether this is their meaning. In addition, we are provided with no evidence to support the putative claim that* VSG121 *can be re-induced at this stage*.

We apologize for this lack of clarity. The observations are now detailed in the text. In fact, re-induction works to the same degree as in non-induced parasites in more than 20 % of the population. The other part revealed different degrees of VSG-induction, which is compatible with gradual reversion of epigenetic silencing, rather than outgrowth of a mutant.

*The authors must comment on this further: if revertants are being selected for that no longer express* VSG121*, it suggests significant problems with the experimental approach*.

The above line of evidence makes it very unlikely that random mutation events are responsible for the decrease of *VSG121* expression.

*A main argument that the authors make to suggest that the* VSG121 *expression is an accurate mimic of VSG switching and differentiation, is that they see a general decrease in transcription, consistent with a previous report by*
[2]*. In more recent work, using whole-genome gene expression analyses (Kabani et al, 2009, BMC Genomics; Capewell et al, 2013, PLoSONE), there is little evidence for such global reduction in transcription: though translation decreases in short stumpy cells relative to long slender, only a very small number of genes show reduced levels of mRNA (e.g., 41 in the Capewell study). Thus, the global effects described here upon* VSG121 *induction may not reflect the developmental change from long slender to short stumpy cells, but instead toxic effects of* VSG121 *overexpression. Thus, the association between VSG switching and differentiation may not be warranted. The authors must discuss these data in the context of their model*.

A) As done by Amiguet-Vercher et al. we analysed the global RNA synthesis rate at single cell resolution by incorporation of BrUTP into nascent transcripts.

B) Kabani et al. and Capewell et al. analysed the mRNA expression profiles of the developmental stages using microarray and digital SAGE approaches, respectively. These techniques involve normalisation to the total RNA input in each sample throughout the analysis, as they aim to identify differentially regulated mRNAs. Thus, these publications contain no information on the overall transcriptional activity or the amount of total RNA in the cells – lower global transcriptional activity or reduced overall RNA amounts would not be detected.

*It is currently not convincing that the* VSG *expression site is a 'global regulator' (end of the Introduction) or 'global control module' (end of the Discussion) of antigenic variation, proliferation and development. Without studies in a fully differentiation-competent 'pleomorphic' strain, strong conclusions can't be drawn. These sections should be toned down to reflect the uncertainty*.

Our observations are indeed based on experiments with monomorphic cells. The fact that developmental competence can be triggered in a strain that is supposed to be partly refractory to developmental progression is mechanistically intriguing. However, we agree that our conclusions would be strengthened by experiments with pleomorphic cells.

Therefore, we have now done the same overexpression in a pleomorphic cell line. Tetracycline-induced expression of a second VSG in fact leads to growth attenuation and the formation of stumpy forms. We have not included these experiments in the present manuscript as they are in part still ongoing and will be published elsewhere. However, we now mention the result in the Discussion.

*2) The strategy used to induce 'switching' involves expression of second VSG, and the authors suggest that such double VSG expression is the trigger for switching – e.g., in the final paragraph of the Introduction “Thus, the VSG itself mediates a feedback loop that controls ... monoallelic gene expression”; “an in situ switch would involve activation of a new ES before silencing of the old one' (first paragraph of the Discussion); and the authors draw parallel with mammalian OR silencing (third paragraph of the Discussion) ”...expression of an ectopic or transgene is sufficient to mediate a feedback inhibition....”. If this is the case, the authors must explain some anomalies. First, this model predicts that two VSGs cannot be simultaneously expressed, as one will trigger suppression of another. However, double* VSG *expressers have been reported, and it has been shown that such double expressers are stable in vitro and in vivo, and do not impair growth: Baltz et al. (1986), Nature; and Munoz-Jordan et al. (1996), Science. How can these data be explained, in the context of the model that the authors propose?*

A) Baltz et al. (1986) describe an observation in *T. equiperdum*, a species that has lost the insect life cycle stage. The report does not include any information on the quantities with which both VSGs are co-expressed. In our model, we simulate a situation in which a second *VSG* ES becomes fully transcriptionally active, resulting in the peak expression of two VSGs at wild type levels. This initial and transient spike in VSG production then quickly triggers a feedback loop that attenuates the active ES. Thus, it is certainly possible to express two *VSG* ES at the same time, albeit not for long and not at the same level. Importantly, Figueiredo et al. (2008) showed that trypanosomes lacking the histone methyltransferase *DOT1B* transcribe two ES simultaneously for up to 40 days. However, at no time were both ES 100 % active. Thus, a situation in which a second ES shows 20 % activity whilst retaining expression from the old one is possible, although such a situation would not be stable in vivo under host immune pressure. This may explain the observations of Baltz et al. (1986) and is absolutely compatible with our model.

B) Munoz-Jordan et al. (1996) generated VSG double-expressors by inserting a second *VSG* into the actively transcribed ES, just upstream of the endogenous *VSG* gene. We have repeated these experiments and carefully quantified the *VSG* levels in several VSG double-expressors and found that both *VSGs* are expressed at a 50/50 ratio, not exceeding a total VSG amount of 100 % when compared to wild type cells. The production of such double-expressor cell lines requires antibiotic selection for 7 days after transfection. During this period the cells must adapt to the production of the two *VSGs* in order to survive. They do this by balancing their expression levels – exactly as our model would predict.

*Second, the authors' model appears to suggest that expression of a second VSG begins a progressive attenuation of the ES from the telomere, allowing the cells to 'probe for ES functionality'. However, how can this actually happen, since in the normal context a second VSG can only be expressed in conjunction with the upstream* ESAGs*, since they share a common promoter in the ES? In other words, it seems that the only circumstances in which* ESAG*s are not co-expressed with the* VSG *is when an ES is activated that lacks these genes; if this is the case, it is only in these circumstances that G1 dormancy and reversible differentiation occurs (these are induced by LOSS of some* ESAGs*, as shown by this work). If a new VSG is expressed from an ES with a full complement of* ESAGs*, these phenotypes would not be seen, further suggesting that they are an artefact of ectopic* VSG121 *expression from outside an ES*.

The reviewer is absolutely right by stating that ES attenuation and cell dormancy would not occur during a ‘regular’ *VSG* switching event. If a new *VSG* ES is activated that contains a full and functional complement of *ESAGs*, none of those would become limiting when the old ES is attenuated. This in situ switch would rapidly be completed without any effects on cell proliferation. This is what we have defined in our model as a 'successful switch' (Figure 8). In contrast, if the newly activated ES does not contain a fully functional or host-compatible set of ESAGs, the trypanosomes would respond to ES attenuation with dormancy and gain of developmental competence. The parasites either adapt to the new ES (as is the case with the unusual *T. b. rhodesiense* SRA-containing ES; see below), they develop in the tsetse vector, or ‘simply’ reactivate the old ES. This is what we have defined as an 'unsuccessful switch' (Figure 8). In our experiments we simulate such an ‘unsuccessful switch’ by expressing a second *VSG* without any *ESAGs*. We discuss this in the Discussion.

*3) Expression site transcription may control developmental competence via at least three genes co-transcribed with the* VSG *but it's difficult to see how each gene knockdown acts redundantly to bring about this effect. Could the authors say more about the putative functions of these* ESAGs *relative to those not linked to this phenotype?*

Most *ESAGs* do not show any significant homologies to other genes present in the genome databases; so it is problematic to speculate about their functions. However, *ESAG1* is predicted to contain a membrane-spanning domain, as well as cytoplasmic endosomal/lysosomal sorting motifs and it apparently localises to the flagellar pocket. This suggests that *ESAG1* might function as a receptor protein (28). Likewise, *ESAG2* is found in the flagellar pocket, but this protein is GPI-anchored and heavily modified with linear p-NAL glycan chains. Thus, *ESAG2* could function in the elaborate endosome pathway (Nolan et al. 1999). It is tempting to speculate that *ESAG1* and 2 act in concert, as this would explain the redundant RNAi phenotypes. In contrast, *ESAG8* is a nucleolar protein that is also distributed in the cytoplasm (Hoek et al. 2000) and interacts with a Puf family member (Hoek et al. 2002). This suggests that *ESAG8* could play a role in the regulation of mRNA stability, maybe during development. *ESAG6/7* form a transferrin receptor required for iron uptake and thus have a crucial function in metabolism. *ESAG3* contains a secretion signal peptide but apparently lacks a membrane-spanning domain and a GPI-anchor (28). We now mention some potential functions in the Discussion.

*Is a SIF signal thought to downregulate these* ESAGs *and in-turn render the cells more competent for differentiation or is ESAG downregulation thought to bypasses SIF signalling?*

We are currently working on the question whether SIF directly induces ES silencing or actively decreases the amount of *ESAG* mRNA. The SIF mediated pathway certainly is the dominant differentiation pathway and involves a complex signaling cascade (Mony et al. 2013). Our results show, however, that ES silencing and concomitant *ESAG* depletion can indeed bypass this pathway.

*4) The* ESAG *sensing model states that* ESAGs 1, 2 *and to some extent* 8 *are important while the other* ESAGs *are not. However,* ESAG 5*, present in all known ES was not tested. The data would be strengthened if* ESAG5 *RNAi was included*.

We had done the *ESAG5* RNAi, but did not observe any phenotype and hence had not include the data. We now mention the result in the Results section entitled “Signaling of expression site attenuation involves three ESAGs.”

*5) The authors’ results quantitating transcripts using hybridisation of dot-blots or Northern blots rather than by qPCR was not completely convincing. In the first Results section the authors say “It is immediately apparent that mRNAs.. decreased faster...” These quantifications should ideally be repeated using qRT-PCR.* ESAG1 *and* ESAG2 *are members of very large gene families, which contain copies outside of ESs, also transcribed by Pol II. Using qPCR, the authors could design primers specific for the* ESAG *copies present in the* VSG221 *ES (and not detect non-ES derived* ESAGs*)*.

We agree that our experiments do not distinguish between ES and non-ES transcripts. As suggested by the reviewer we have now tried qPCR, however have not been able to unequivocally distinguish between the ES-encoded and non-ES mRNA species. In fact, the *ESAG1* and *2* family members display very high sequence homologies, which may explain this result.

The percentage of non-ES derived *ESAG1* transcripts was reported to be at most 20 % of total *ESAG1* mRNA (7). This means that the vast majority of *ESAG* transcripts is generated by the active ES. Thus, a decrease in mRNA abundance of more than 20 % must be due to reduced transcripts from the ES. We measured a decrease of *ESAG1* and *2* mRNA of 60 % and 40 %, respectively.

Importantly, *ESAG8* is specific for the ES. Therefore, the *ESAG8* knock-down only affects the ES-linked gene.

[Editors' note: further clarifications were requested prior to acceptance, as described below.]

*1) One point in particular remains unclear, which is an important issue: the authors say that “it is certainly possible to express two VSG ES at the same time, albeit not for long and not at the same level” and then that “cells must adapt to the production of the two VSGs in order to survive. They do this by balancing their expression levels – exactly as our model would predict”. This seems contradictory. Would the model not predict attenuation of the downstream* VSG *in the experiments reported by Munoz-Jordan et al. (1996)?*

This is correct. We actually had done the experiment but did not mention the results. The paragraph beginning “*“This model further explains the expression levels in stable VSG double-expressors…”* is now included in the Discussion.

*2) In the absence of knowing the functions of ESAGs 1, 2 and 8, the evidence that ESAGs1, 2 and 8 serve as signals for ES attenuation still a bit speculative. I think that the Abstract should read that these “appear to serve” or “could serve” as a signal for ES attenuation*.

We have changed the Abstract accordingly.